# Alpha-Lipoic Acid Reduces Neuroinflammation and Oxidative Stress Induced by Dapsone in an Animal Model

**DOI:** 10.3390/nu17050791

**Published:** 2025-02-25

**Authors:** Bruno Alexandre Quadros Gomes, Savio Monteiro dos Santos, Lucas da Silva Gato, Kaio Murilo Monteiro Espíndola, Rana Karen Mesquita da Silva, Kelly Davis, Kely Campos Navegantes-Lima, Rommel Mario Rodriguez Burbano, Pedro Roosevelt Torres Romao, Michael D. Coleman, Marta Chagas Monteiro

**Affiliations:** 1Postgraduate Program in Neuroscience and Cell Biology, Federal University of Pará/UFPA, Rua Augusto Corrêa 01, Bairro Guamá, Belém 66075-110, PA, Brazil; bhrunoquadros10@gmail.com (B.A.Q.G.); saviomontsan@gmail.com (S.M.d.S.); kaioespindola@hotmail.com (K.M.M.E.); rannamesquita184@gmail.com (R.K.M.d.S.); 2Postgraduate Program in Pharmaceutical Sciences, Faculty of Pharmacy, Federal University of Pará/UFPA, Rua Augusto Corrêa 01, Bairro Guamá, Belém 66075-110, PA, Brazil; kcnavegantes@gmail.com; 3Laboratory Immunology, Microbiology and In Vitro Assays (LABEIM), Faculty of Pharmacy, Federal University of Pará/UFPA, Belém 66075-110, PA, Brazil; lucas.sg1789@gmail.com; 4Postgraduate Program in Pharmacology and Biochemistry, Faculty of Pharmacy, Federal University of Pará/UFPA, Rua Augusto Corrêa 01, Bairro Guamá, Belém 66075-110, PA, Brazil; kelly87davis@gmail.com; 5Laboratory of Molecular Biology, Ophir Loyola Hospital, Belém 66063-240, PA, Brazil; rommel@ufpa.br; 6Laboratory of Cellular and Molecular Immunology, Federal University of Health Sciences of Porto Alegre, Porto Alegre 90050-170, RS, Brazil; pedror@ufcspa.edu.br; 7College of Health and Life Sciences, Aston University, Aston Triangle, Birmingham B4 7ET, UK; m.d.coleman@aston.ac.uk

**Keywords:** dapsone, neuroinflammation, alpha lipoic acid

## Abstract

**Background/Objectives:** Chronic treatment with dapsone (DDS) has been linked to adverse reactions involving all organ systems, such as dapsone hypersensitivity syndrome, methemoglobinemia and hemolytic anemia, besides neuroinflammation and neurodegeneration due to iron accumulation and oxidative stress. These effects probably occur due to the presence of its toxic metabolite DDS-NOH, which can generate reactive oxygen species (ROS) and iron overload. In this sense, antioxidant compounds with chelating properties, such as alpha-lipoic acid (ALA), may be an interesting adjuvant therapy strategy in treating or preventing these effects. Thus, the aim of this study was to evaluate the effects of ALA on oxidative and neuroinflammatory changes caused by DDS treatment in the prefrontal cortex and hippocampus of mice. **Materials and Methods:**
*Mus musculus* male mice that were pre-treated with DDS (40 mg/kg) and post-treated with ALA (25 mg/kg) underwent analyses for oxidative stress, antioxidant capacity, cytokine expression and microglial/astrocytic activity. **Results:** DDS did not activate macrophages/microglia or astrocytes in the prefrontal cortex but induced their activation in the hippocampus. ALA stimulated a protective microglial profile and reduced astrocyte reactivity, especially in the hippocampus. DDS increased the pro-inflammatory cytokine IL-1β and reduced brain-derived neurotrophic factor (BDNF), effects reversed by ALA. DDS also reduced antioxidant capacity (TEAC, GSH, SOD, CAT) and increased oxidative damage (lipid peroxidation, iron accumulation), while ALA restored antioxidant levels and reduced oxidative stress. **Conclusions:** ALA was able to reduce the effects of DDS, such as reducing microglial and astrocytic activation, as well as to decrease the levels of pro-inflammatory cytokines and increase BDNF, in addition to increasing antioxidant capacity and reducing oxidative damage caused by iron accumulation. Therefore, ALA is considered a useful and promising therapeutic alternative for the treatment of diseases related to oxidative stress and neuroinflammation.

## 1. Introduction

Dapsone (DDS) is a synthetic derivative of sulfones, with antimicrobial activities, which is used in the treatment of autoimmune diseases such as bullous systemic lupus erythematosus [1]. Regarding adverse reactions triggered by using DDS, symptoms such as erythroderma, hemolytic anemia, jaundice, methemoglobinemia and sulfone syndrome may be cited, occurring in the presence of rash, fever and eosinophilia [2]. In addition, degenerative disorders in the central nervous system (CNS) can be associated with iron accumulation, due to disorders in its metabolism, that can contribute to the process of oxidative stress, inducing inflammation and neuronal damage [3,4].

Neuroinflammation triggered by this process involves the activation of microglia, which contributes to the release of pro-inflammatory factors, including cytokines, reactive oxygen species (ROS), reactive nitrogen species (RNS) and eicosanoids, leading to damaged neurons and glial cells, which contribute to neurodegenerative disorders and cognitive impairment [5,6,7]. These inflammatory processes trigger a series of events, including the increased production of ROS and RNS, the homeostatic imbalance of iron metabolism and mitochondrial dysfunction [8,9]. In this context, two areas of the CNS are particularly interesting, the prefrontal cortex (PFC) and the hippocampus [10,11]. The hippocampus is one of the most sensitive areas of the CNS, mainly associated with memory formation, as well as emotional, adaptive and reproductive behaviors [12,13]. Metabolic disorders in the hippocampus can be caused by oxidative stress and iron accumulation [13,14]. The PFC controls important functions such as emotion, memory, language, distraction inhibition and decision making [15,16,17]. Oxidative changes in the PFC are associated with neurodegeneration and the pathophysiology of neuropsychiatric diseases, such as anxiety and depression [18,19].

Antioxidants are crucial for improving the redox status of brain cells in many neurodegenerative diseases involving memory loss, impaired cognitive function, neuronal apoptosis, neuroinflammation and metal-induced neurodegeneration [20,21,22,23]. Alpha lipoic acid (ALA or 1,2-dithiolane-3-pentanoic acid) is a natural compound with antioxidant, anti-inflammatory and metal-chelating properties, which acts as an enzymatic cofactor and is able to regulate metabolism, energy production and mitochondrial biogenesis [24,25,26,27]. ALA can easily penetrate the blood brain barrier (BBB) and is reduced by an NADH-dependent reaction, forming dihydrolipoic acid (DHLA), the oxidized and reduced forms of which are both potent antioxidants [28,29]. In this regard, evidence indicates that ALA delays the aging process in the brain, improving brain function and memory [26,30,31].

In human studies, ALA has been shown to be effective in slowing cognitive decline in patients with Alzheimer’s disease [32,33]. The beneficial effects of ALA on neural oxidative homeostasis enables its application as a therapeutic agent for treating various neurological disorders such as Alzheimer´s disease [25], depression [34], schizophrenia [35], Parkinson’s disease [36] and cognitive dysfunction [37,38].

In this context, ALA becomes an interesting therapeutic strategy, due to its antioxidant properties, acting to reduce the production and release of ROS, oxidative changes and neurotoxicity in the CNS [39], as well as presenting chelating properties [40]. Therefore, this work aims to evaluate the oxidative and neuroinflammatory changes caused by DDS in the hippocampus and PFC in mice, as well as the possible antioxidant, anti-inflammatory and immunomodulatory effects of ALA on the changes caused by DDS.

## 2. Materials and Methods

### 2.1. Chemicals

Lipoic acid (ALA), methanol, ethanol, dimethyl sulfoxide, sodium hydroxide, sodium chloride, ethylenediamine tetraacetic acid (EDTA), hydrogen peroxide (H_2_O_2_), hypoxanthine, ethidium bromide, xanthine oxidase and cytochrome C were purchased from Sigma Chemical Com. (St. Louis, MO, USA). Dapsone (DDS) was purchased from Santa Cruz Biotechnology (Santa Cruz, CA, USA), DMEN-F12 with ʟ-glutamine, HEPES, sodium pyruvate and sodium bicarbonate, liquid, sterile-filtered, suitable for cell culture, Sigma Chemical Com. (St. Louis, MO, USA). Fetal Bovine Serum (FBS, Sigma Chemical Com. St. Louis, MO, USA).

### 2.2. Ethics Statement

This study was carried out in accordance with the recommendations of the Guide for the Care and Use of Laboratory Animals of the Brazilian National Council for Animal Experimentation (http://www.sbcal.org.br/) and the NIH (Guidelines for the Care and Use of Laboratory Animals). The Institutional Animal Ethics Committee of the University of Pará/UFPA (CEUA, Protocol: 2411100816) approved all procedures used in this study in 28 October 2017.

### 2.3. Mice

In the experiments, we included a total of 60 *Mus musculus* male mice, Swiss Webster, young adults and healthy (8 weeks old, 35–40 g), obtained from the Evandro Chagas Institute (IEC-PA). Animals outside these criteria were excluded from the study. Animals were kept in cages (size 30 × 20 × 13 cm; in polypropylene), 5 animals per cage, under controlled conditions of temperature (25 ± 1 °C) and light (12 h light/dark cycle), with food (Supralab^®^, Alisul, Brazil) and water ad libitum and acclimatized to conditions for 3 days before use.

### 2.4. Administration of Dapsone and ALA

Dapsone was dissolved in 2% dimethyl sulfoxide (DMSO) and administered intraperitoneally (i.p.) at a concentration of 40 mg/kg, according to the methodology described by Bergamaschi, with adaptations [41]. ALA was dissolved in 0.9% saline solution and administered orally (gavage) at a concentration of 25 mg/kg, based on the methodology described by [42,43], with adaptations in dose and treatment time, to evaluate its effects on acute exposure and toxic doses of DDS.

The mice were divided randomly into 3 groups (*n* = 20 animals/group) according to the treatment schedule: (i) control group (animals treated with DMSO 2% and post-treated with 0.9% saline solution), (ii) DDS (animals treated with DDS (40 mg/kg) and post-treated with 0.9% saline solution) and (iii) ALA (animals treated with DDS and post-treated with ALA at dose of 25 mg/kg, given 2 h after DDS). Ten animals from each group were used for flow cytometry, and ten were used for the evaluation of cytokines and oxidative parameters. Only the main researcher knew which group the cages with animals belonged to and the different stages of the experiment. All treatments were carried out for 5 consecutive days, following the order of the groups [44]. In the experiments, within 4 h after treatment, all animals from each group were euthanized with ketamine 80 mg/kg and xylazine 10 mg/kg IP, followed by cardiac puncture to obtain blood (Figure 1). No unexpected adverse events were observed during the experiments. This study did not have humane endpoints. The PFC and hippocampus were collected and stored at −80 °C for the tissue disruption process to obtain the homogenate.

### 2.5. Determination of Trolox Equivalent Antioxidant Capacity (TEAC)

The total antioxidant capacity of samples from mice was evaluated by Trolox equivalent antioxidant capacity assay (6-Hydroxy2,5,7,8-tetramethyl chroman-2-carboxylic acid; Sigma-Aldrich, St. Louis, MO, USA). In this assay, 2,2′Azino-bis(3-ethylbenzothiazoline-6-sulfonic acid)diammonium salt (ABTS) (Sigma Aldrich) was incubated with potassium persulfate (Sigma Aldrich) to produce ABTS•^+^, a green/blue chromophore. The inhibition of ABTS•^+^ formation by antioxidants in the samples was expressed as Trolox equivalents, determined at 740 nm [45].

### 2.6. Determination of Superoxide Dismutase Activity

SOD activity was evaluated in samples from mice, through the indirect detection of this enzyme, which promotes 50% inhibition of the reduction of cytochrome C at 25 °C, pH 7.8, since SOD promotes the conversion of superoxide anion (O_2_^•−^) into H_2_O_2_ and O_2_, preventing the reduction of cytochrome C, as detected by spectrophotometry at 550 nm [46].

### 2.7. Determination of Catalase Activity

Catalase enzyme activity was determined according to the method described by Aebi [47], which evaluates the capacity of the catalase present in the sample to convert H_2_O_2_ (Merck) into water and oxygen. Thus, to assess the decay of H_2_O_2_, aliquots of the diluted samples were added to 900 μL of reaction solution (TRIS base buffer, H_2_O_2_ 30% and ultrapure water) at pH 8. The decrease in H_2_O_2_ concentration was checked at λ = 240 nm in a spectrophotometer at 25 °C for 60 s. Catalase activity was defined as the activity required to degrade 1 mol H_2_O_2_ in 60 s, at pH 8 at 25 °C and was expressed as U/mg protein in the tissues. The enzyme activity data obtained were normalized by the respective total protein concentrations, using the commercial Doles Kit (Goiânia, Brazil) in spectrophotometer (Biospectro, SP-220 model).

### 2.8. Determination of Reduced Glutathione Activity

The determination of GSH concentrations was performed in samples from animals, based on the ability of this antioxidant to reduce 5,5-dithiobis-2-nitrobenzoic acid (DTNB) (Sigma-Aldrich) to 5-thio-2-nitrobenzoic acid (TNB), which was quantified by spectrophotometry at 412 nm [48].

### 2.9. Determination of Lipid Peroxidation

Lipid peroxidation was measured in samples from mice, as an indicator of oxidative stress, using the thiobarbituric acid reactive substance (TBARS) assay, where the reaction of MDA and other substances with thiobarbituric acid (TBA; Sigma-Aldrich), at pH 2.5 and 94 °C, forms the pink colored MDA-TBA complex. The reading was performed in a spectrophotometer at an absorbance of 535 nm [49].

### 2.10. Determination of Iron Concentration

Iron levels were measured in PFC and hippocampus using the modified Goodwin colorimetric method (Labtest^®^ kit, Lagoa Santa-Minas Gerais, Brazil), which was measured in a spectrophotometer at 560 nm (Biospectro, SP-220 model).

### 2.11. Flow Cytometry

The PFC and hippocampus from basal animals and animals treated with DDS or DDS+ALA (*n* = 10 per each group) were collected and mechanically dissociated individually in 10 mL of complete medium DMEM/F12. Cellular debris was removed by passing the cell suspension through a 70-μm cell strainer (Falcon™, Gibco, Thermo Fisher Scientific, Waltham, MA, USA). The samples were then centrifuged at 1200 RPM for 15 min, the supernatant was discarded, and the pellet was resuspended in sterile 1X PBS, followed by another centrifugation at 1200 RPM for 15 min. Next, the sample was treated with Red Blood Cell (RBC) lysis buffer (8.3 g NH₄Cl, 1.0 g KHCO_3_, 1.8 mL of 5% EDTA and distilled H_2_O to volume), centrifuged again at 1200 RPM for 15 min, and the supernatant was discarded. The pellet was then resuspended in complete DMEM/F12 medium. Finally, the cells were counted, and the concentration was adjusted to obtain 1.5 × 10^8^ cells/mL [50,51] A total of 1.5 × 10^8^ cells/mL were pre-incubated for 15 min at room temperature with 5 µL of one of the following monoclonal antibodies: anti-F4/80 conjugated to PE (Sigma-Aldrich) or anti-GFAP conjugated to FITC (Thermo Fisher Scientific). The samples were then centrifuged at 450× *g* for 7 min at 4 °C. The plate was washed with 1X PBS, and the wells were resuspended in 50 µL of buffer containing the respective antibodies (PE–TLR-4/FITC–F4/80) at a 1:200 dilution. Flow cytometry analysis was performed using a FACSVerse flow cytometer and FlowJo software v10 (Becton, Dickinson and Company, Franklin Lakes, NJ, USA). The final results were expressed as median fluorescence intensity (MFI).

### 2.12. Determination of IL-1β, IL-17, IL-4 and BDNF Expression

Cytokines IL-1β, IL-17 and IL-4, as well as BDNF, were assessed in the homogenized PFC and hippocampus of the mice using commercial kits (R&D Systems, Billings, MT, USA) by Enzyme-Linked Immunosorbent Assay (ELISA) method, with a detection limit for each cytokine of 5 pg/mL.

### 2.13. Statistical Analysis

Statistical analyses were performed using GraphPad Prism 8 software (GraphPad Software Inc., La Jolla, CA, USA). The data obtained were analyzed by analysis of variance (ANOVA), followed by Tukey’s test for multiple comparisons. Results were expressed as mean ± standard deviation and considered statistically significant for *p* ≤ 0.05.

## 3. Results

### 3.1. Effects of DDS and ALA on the Cellular Profile and Cytokines in the Prefrontal Cortex and Hippocampus

Animals treated with DDS (40 mg/kg) showed an increase in the percentage of the macrophage/microglial population that expressed low levels of the F4/80+ marker in the PFC and hippocampus (Figure 2A,C respectively), when compared to the control group. The indices of the activation or differentiation of macrophages/microglia following *high expression* were not altered in the PFC but were increased in the hippocampus. This shows that DDS did not activate macrophage/microglial cells in the PFC but was able to activate these cells in the hippocampus. In relation to post-treatment with ALA (25 mg/kg), the activation or differentiation of macrophages/microglia following *high expression* was observed in the PFC but reduced in the hippocampus (Figure 2A,C).

Regarding astrocytes in the PFC, treatment with DDS (40 mg/kg) increased the percentage of astrocytes with low expression of glial fibrillary acidic protein (GFAP) in relation to the control (Figure 2B). However, the percentage of astrocytes activated following high expression decreased. On the other hand, post-treatment with ALA (25 mg/kg) increased the percentage of the GFAP+low population and decreased the GFAP+high population in relation to the DDS group, showing an immunoregulatory effect on astrocytes induced by DDS (Figure 2B). In this context, high and low represent the expression of the cellular marker, allowing for a quantitative analysis of cells. In the astrocytes and microglia evaluated in our study, it was also possible to estimate the percentage of cells present, as well as those that were activated or inactive in the tissue and expressed the marker.

In the hippocampus, DDS decreased the percentage of astrocytes showing the low expression of GFAP and increased the percentage of astrocytes with high expression of GFAP compared to the control group (Figure 2D). Post-treatment with ALA (25 mg/kg) showed an immunoregulatory effect on DDS-stimulated astrocytes, returning the profile of astrocytes with low and high GFAP expression to baseline levels (Figure 2D).

The animals treated with DDS 40 mg/kg showed an increased expression of IL-1β, IL-17 and IL-4 in the PFC and hippocampus in relation to the control group; however, the treatment reduced the production of BDNF, showing that DDS can induce an inflammatory process in the PFC (Figure 3A,B,D) and the hippocampus (Figure 4A,B,D). Post-treatment with ALA (25 mg/kg) was able to reduce IL-1β production and increase BDNF levels in the PFC and hippocampus but did not change IL-17 and IL-4 levels after treatment with DDS, showing the anti-inflammatory and neuroprotective effects of ALA in the PFC (Figure 3A–C) and hippocampus (Figure 4A–C).

### 3.2. Effects of DDS and ALA on Antioxidant Capacity and Oxidative Stress in the Prefrontal Cortex and Hippocampus

Regarding antioxidant parameters, the effects of DDS on TEAC, GSH, SOD and CAT concentrations in the PFC and hippocampus were evaluated. Our results showed that DDS decreased antioxidant capacity in all of the analyzed parameters (Figure 5A–H). However, ALA (25 mg/kg) was able to inhibit the oxidative action of DDS and increase or restore the levels of TEAC, GSH, SOD and CAT in the PFC for samples treated with DDS. In the hippocampus, ALA (25 mg/kg) presented a similar characteristic to that in the PFC, being able to reverse DDS oxidation in TEAC, GSH CAT (Figure 5E,F,H); however, it did not restore SOD concentrations (Figure 5G).

In addition, TBARS and iron levels were evaluated in animals treated with DDS (40 mg/kg). The results show increased levels of TBARS and iron induced by DDS in the PFC (Figure 6A,B) and hippocampus (Figure 6C,D), while ALA (25 mg/kg) was capable of inhibiting lipid peroxidation and iron accumulation (Figure 6A–D).

## 4. Discussion

The low molecular weight and lipid solubility of DDS contribute to the fact that the sulfone diffuses easily into the CNS and increases the ability to cause oxidative damage to nerve cells and tissues. Therefore, to evaluate the neuroinflammatory activity of DDS, as well as the role of ALA in this process, we quantified the expression of neuroglial markers, such as F4/80+ expression, in relation to macrophage/microglial activity, and GFAP expression, a marker that is related to astrocytic activity in the PFC and hippocampus of mice. Our data showed that treatment with DDS (40 mg/kg) did not lead to macrophage/microglia activation and did not stimulate astrocyte reactivity in the PFC but activated macrophage/microglial cells and astrocytes in the hippocampus of mice. On the other hand, post-treatment with ALA (25 mg/kg) stimulated the activation of a macrophage/microglia profile but reduced astrocyte reactivity in the PFC. In the hippocampus, ALA reduced macrophage/microglia and astrocyte activation, showing an important immunomodulatory effect.

These results showed that DDS initially induced macrophage/microglia and astrocyte influx in the PFC but did not lead to the activation of these cells; however, ALA led to macrophage/microglia activation and reduced astrocyte reactivity, showing an immunostimulatory action on macrophage cells, which is fundamental for the regulation of ROS production, oxidative stress and neurotoxicity in these tissues. In the hippocampus, DDS first reduces the influx of macrophage/microglial cells and astrocytes but leads to activation. On the other hand, ALA reduces this activation, showing an important immunomodulatory effect in the hippocampus. According to Kim et al. [52], treatment with ALA interferes with microglial activation, reducing this effect, while also modulating the macrophage profile, inhibiting the M1 (inflammatory) macrophage phenotype and stimulating the M2 (anti-inflammatory) profile, which possibly reduces neuronal damage. Wang et al. [53], in a rat model of middle cerebral artery occlusion and an in vitro model of LPS-induced microglial inflammation, showed that ALA (20 mg/kg and 40 mg/kg) decreased infarct size, cerebral edema and neurologic deficits. Furthermore, ALA induced microglia polarization to the M2 phenotype, modulated the expression of IL-1β, IL-6, TNF-α and IL-10 and attenuated the activation of NF-κB. This suggests that ALA has a beneficial effect on experimental stroke, via modulation of M1/M2 microglia polarization, with the potential mechanism of ALA being the inhibition of NF-κB activation.

The processes related to memory maintenance and learning are related to synaptic modulation, which can be regulated by neurotrophins, which, in turn, also act on the long-term potentiation of the hippocampus. In the brain, ALA acts to increase BDNF levels [54] through the activation of tyrosine kinase B (TrkB) and NF-κB receptors, and the action of BDNF is related to neurogenesis, neuroprotection and neuroplasticity, due to the promotion of myelination, differentiation and the proliferation of oligodendrocyte precursor cells [55]. Zhang et al. [56] found that ALA improves cognitive function in aging mice, and its protective mechanisms occur through the modulation of synaptic transmission, lipid transporter activity and neuroinflammation mechanisms.

In addition to microglial and astrocytic activation, in the context of neuroinflammation and anti-inflammatory activity, we also evaluated the expression of pro-inflammatory (IL-1β and IL-17) and anti-inflammatory (IL-4) cytokines, as well as BDNF production in the PFC and hippocampus. We observed that treatment with DDS (40 mg/kg) induced the expression of IL-1β, IL-17 and IL-4 but reduced the production of BDNF, which suggests that DDS induces an inflammatory and oxidative process in these tissues and can even lead to anxiety and depression in animals by negatively interfering with BDNF signaling pathways. Interestingly, DDS did not promote the activation of microglia and astrocytes, but it is important to consider other factors involved in the context of neuroinflammation in the PFC and hippocampus, such as pro-inflammatory cytokines and oxidative stress. Thus, it is likely that DDS may be involved in neuroinflammation through mechanisms that are still poorly understood but depend on the activation of pro-inflammatory cytokines, reduced antioxidant capacity, ROS production and iron concentrations.

Li et al. [57] reported the inflammatory profile of microglia in mice with different anxiety characteristics and found a high anxiety profile in DBA/2J naïve mice and significant microglial M1 polarization. Furthermore, mice expressed higher levels of IL-1β, IL-6 and TNF-α mRNA in the hypothalamus before and after LPS stimulation. However, post-treatment with ALA (25 mg/kg) was able to reduce IL-1β production and increase BDNF levels but did not alter IL-17 and IL-4 levels in either tissue, showing its important anti-inflammatory and antioxidant role. Furthermore, in a study developed by Dinicola et al. [58], it was possible to verify the epigenetic regulatory activity that ALA can induce on the expression of IL-1β [59].

In view of this, the current study found that the mRNA levels of the two genes in the ALA-treated group were lower compared to the untreated group. Regarding the identification and quantification of IL-1β and IL-6, these were performed by the ELISA method, where lower levels of supernatant were perceptible in the culture of cells that received treatment with ALA compared to those that did not receive such treatment. In addition, the levels of mRNA encoding interleukins in the treated cell cultures were inversely proportional to DNA methylation. Therefore, with such data, it is possible to infer that there is modulation by ALA, directly or indirectly, of the pro-inflammatory cytokines related to processes such as neurodegeneration [58]. Giustina et al. [60] showed that ALA reduced BBB permeability and TNF-α levels in the hippocampus within 24 h, as well as IL-1β levels in the hippocampus and PFC. Furthermore, ALA protected BBB permeability, decreased neuroinflammation by reducing TNF-α and IL-1β levels and acutely increased antioxidant activity in brain structures after sepsis induction.

Furthermore, we evaluated the effects of DDS on redox state and neuroinflammation in the PFC and hippocampus. In our experimental model, treatment with DDS (40 mg/kg) decreased TEAC, SGH, SOD and CAT in both tissues. These results suggest that DDS and its metabolites can cross the BBB and reduce brain antioxidant status, promoting oxidative stress. The in vitro DDS-induced toxicity models showed that the presence of pro-inflammatory cytokines was related to increased DDS and DDS-NOH toxicity through a reduction in GSH levels. In fact, the effects of pre-treatment with TNF-α on GSH and ROS levels were evaluated, and a significant decrease in GSH levels was observed after 24 h, while ROS formation was increased in the presence of DDS-NOH [61]. Therefore, it is likely that DDS and its metabolites can cause neuroinflammation and neurotoxicity, as well as oxidative stress in the CNS, in response to the production and effect of cytokines.

Our results also demonstrated that treatment with ALA (25 mg/kg) increased or restored the levels of TEAC, SGH, SOD and CAT in the PFC after treatment with DDS. In the hippocampus, ALA (25 mg/kg) showed a similar behavior in the PFC, being able to increase or restore the levels of TEAC, GSH and CAT, but did not increase the concentrations of SOD. In this way, ALA can act on the CNS, restoring antioxidant capacity, and can be a viable adjuvant alternative for patients who are chronic users of DDS. Our data corroborate those of Al Shahrani et al. [62], who found that ROS can cause mitochondrial dysfunction and reduce GSH levels in brain tissue, resulting in oxidative stress. Bilska et al. [39] also showed that ALA treatment increased GSH concentrations without any changes in GSSG and decreased NO concentrations, attenuating oxidative stress induced by reserpine in rats.

Tanbek et al. [63] evaluated the effects of ALA (100 mg/kg/day) against brain tissue (hippocampus, cortex, hypothalamus and striatum) damage caused by diabetes and found increased activities of SOD, CAT, GSH-Px and reduced MDA levels in rats with diabetes. Thus, restoring cell damage and cognitive functions in brain tissue had neuroprotective effects without showing any antidiabetic effects. In the ALA-treated group, CAT levels were similar to those in the control group; on the other hand, decreases in SOD values were noted. Thus, it can be seen that ALA has benefits against oxidative stress with regard to diseases associated with neuronal damage [64]. At the same time, studies which evaluated a profile aimed at neuroinflammation suggest ALA doses of 100 to 200 mg/kg to inhibit inflammatory pathways in the central nervous system, such as the mitogen activated protein kinase (MAPK) signaling pathway, related to the ERK, JNK and NF-kB pathways. Under conditions of metabolic disorders, the cellular microenvironment can accumulate ROS and thus modify it, consequently activating signaling pathways for inflammatory processes [15]. Therefore, it is likely that ALA can act directly as an inducer of GSH synthesis in the PFC and also indirectly through the SIRT-1 pathway to increase antioxidant capacity and reduce the toxic effects of DDS and MetHb. In fact, studies report that stress promotes changes in ROS and GSH levels, as well as the activity of antioxidant enzymes in brain regions such as the PFC and hippocampus [65,66,67].

Furthermore, neuroinflammation in the hippocampus region, triggered by DDS and its metabolites (DDS-NOH), is possibly related to the decrease in GSH and consequent increase in ROS, corroborating oxidative stress, which causes changes in the neuronal environment that can result in inflammatory processes. In addition, DDS and DDS-NOH are capable of causing hemoglobin oxidation and hemolysis, resulting in the release of iron into the bloodstream, in order to contribute to oxidative manifestations in various organs including the brain in the same potency in mice and humans. In this context, iron overload is capable of triggering the production of pro-inflammatory cytokines, such as TNF-α, IL-1β and INF-γ, which contribute to the formation of ROS and RNS, both of which are capable of causing neurotoxicity and death from neuronal disorders, which are often reported in cases of neurodegenerative diseases [3].

Increased iron levels accompanied by cerebral oxidative stress are associated with neurodegenerative diseases, so the abnormal increase in iron in brain tissue can lead to the exacerbated generation of ROS, inducing neurotoxicity and neuronal death [68]. In this sense, the characteristics of iron and its homeostasis are linked to the inflammatory response and infection; therefore, these are important pathways that can explain susceptibility to disease and the response to infection and inflammation [69].

We also evaluated TBARS and iron levels in animals treated with DDS (40 mg/kg). The results show increased levels of TBARS and iron induced by DDS in the PFC (Figure 5A,B) and hippocampus (Figure 5C,D), while ALA was capable of inhibiting or reducing the lipid peroxidation and iron accumulation (Figure 5A–D), which may suggest that the PFC and hippocampus are sensitive to oxidative stress. Therefore, DDS can alter the redox balance, promoting oxidative stress and inducing the synthesis and release of MDA in the PFC. This can lead to neurotoxicity and neuroinflammation, which may explain the clinical changes observed in patients with high levels of MetHb. These effects may occur indirectly through peripheral oxidative stress, involving the release of iron and oxidation of MetHb. Piloni et al. [70] administered Fe-dextran intraperitoneally to study the oxidative stress triggered by Fe overload in the brains of rats and found a significant increase in the content of total Fe and free intracellular iron in the cortex (2.4 times), hippocampus (1.6 times) and striatum (2.9 times) 6 h after Fe administration, suggesting that this is a crucial event in oxidative stress in the brain. According to a study by Reddy and Labhasetwar [71], iron overload increases MDA levels and reduces SOD expression.

The dysregulation of iron metabolism forms hydroxyl radicals through the Fenton reaction, triggering oxidative stress reactions and damaging lipids, protein and DNA structures and functions in cells [72]. Iron accumulation in the hippocampus is associated with the development of neurodegenerative diseases such as Alzheimer’s disease (AD) due to oxidative damage and increased amyloid-beta aggregation [73]. Jahanshahi et al. [74] showed that iron-chelating antioxidants reduced the formation of amyloid-beta plaques in the hippocampi of male mice, thus possibly aiding in the treatment of AD. Studies with ALA demonstrate neuroprotective effects by suppressing brain oxidative stress [75,76], reducing cognitive impairment [77,78], restoring neuronal and neurotransmitter functions [31,37,79], reducing neuronal apoptosis [38], alleviating neuronal degeneration, inhibiting ferroptosis [40], reversing low concentrations of brain-derived neurotrophic factor (BDNF) in the hippocampi of animal models [34], regulating iron homeostasis and reducing oxidative stress levels in nigral neurons [36].

In the CNS, oxidative stress induced by DDS treatment leads to tissue damage. Although the mechanisms are not yet clearly understood, CNS damage appears to be related to oxidized iron during the formation of methemoglobin [80]. Our data show toxicity caused by DDS in an animal model due to iron accumulation in the hippocampus, reinforcing evidence of the neurotoxic and oxidative potential of this drug. Thus, the increase in accumulated iron levels in the hippocampus regions of animals treated with DDS, as observed in this study, shows that the use of DDS can alter regulatory mechanisms of iron transport in the CNS, thereby triggering processes that lead to cellular and tissue damage and culminating in pathophysiological effects in the CNS. On the other hand, post-treatment with ALA showed lower levels of iron, suggesting a regulatory mechanism for this compound.

However, it is likely that ALA crosses the BBB and/or acts indirectly via cell signaling, effecting the PFC by reducing ROS and oxidative aggression caused by DDS. Della Giustina et al. [50] also found that ALA has an important antioxidant effect on brain dysfunction in rats, especially in the PFC and hippocampus, by decreasing myeloperoxidase (MPO) and NO activity, lipid and protein peroxidation, increasing antioxidant capacity and CAT activity and preventing cognitive dysfunction. Other ALA mechanisms have been proposed, such as those reported by Chen et al. [81], who showed the effects of ALA on the expression of the transferrin receptor TfR1, DMT1, Fpn1 and ferritin in BV-2 microglial cells. Their data showed that ALA can increase the expression of TfR1, DMT1 and ferritin, causing an increase in iron outside the cell, promoting the deposition of iron in ferritin, increasing cellular iron absorption and reducing free iron both inside and outside the cell. According to Xie et al. [82], ALA can attenuate the damage caused by ROS and reduce neuroinflammatory activities; therefore, ALA may have a positive effect on neuronal ferroptosis, an iron- and ROS-dependent form of cell death. Zhang [41] also reported that ALA supplementation in mice can potentially inhibit caspase-dependent neuronal apoptosis. Furthermore, ALA blocked Tau-induced iron overload and ferroptosis, lipid peroxidation and neuroinflammation.

Therefore, the possible therapeutic use of ALA depends on its antioxidant properties, the ability to remove or prevent the formation of ROS and its chelating properties, enabling it to remove excess iron from the CNS, as well as its immunomodulatory properties. Thus, ALA must essentially be able to cross cell membranes, as well as the BBB, to combat ROS generated during oxidative stress, reaching regions with iron accumulation, without reducing iron bound to plasma transferrin and being able to remove iron from the site of accumulation or transfer it to other biological proteins such as circulating transferrin. Therefore, we believe that ALA is a useful and promising therapeutic alternative for the treatment of diseases and conditions related to increased oxidative stress and neuroinflammation, especially in the PFC and hippocampus, during the chronic use of DDS, by reducing adverse reactions and increasing adherence to treatment.

## 5. Conclusions

From this study, we concluded that dapsone reduces antioxidant capacity in the PFC and hippocampus. In addition, evidence of neuroinflammation, such as microglial and astrocyte influx, the elevation of inflammatory cytokines such as IL-1β, IL-17 and IL-4 and the reduction of BDNF, as well a higher iron concentration, can be attributed to the effects of DDS on the CNS. On the other hand, post-treatment with ALA stimulated macrophage activation and inhibited astrocyte reactivity, as well as inhibiting IL-1β production and increasing BDNF in the PFC and hippocampus. ALA showed the ability to reverse the inflammatory profile, with a reduction in markers such as IL-1β and BDNF, as well as re-establishing antioxidant capacity, with increased levels of SOD, CAT, GSH and TEAC and reduced lipid peroxidation and iron levels, both in the PFC and hippocampus. These results, together with future research into the potential uses of ALA as a neuroprotective agent, could bring various benefits to people who use oxidizing drugs such as DDS, as well as to those with other diseases whose pathogenesis is inflammatory.

## Figures and Tables

**Figure 1 nutrients-17-00791-f001:**
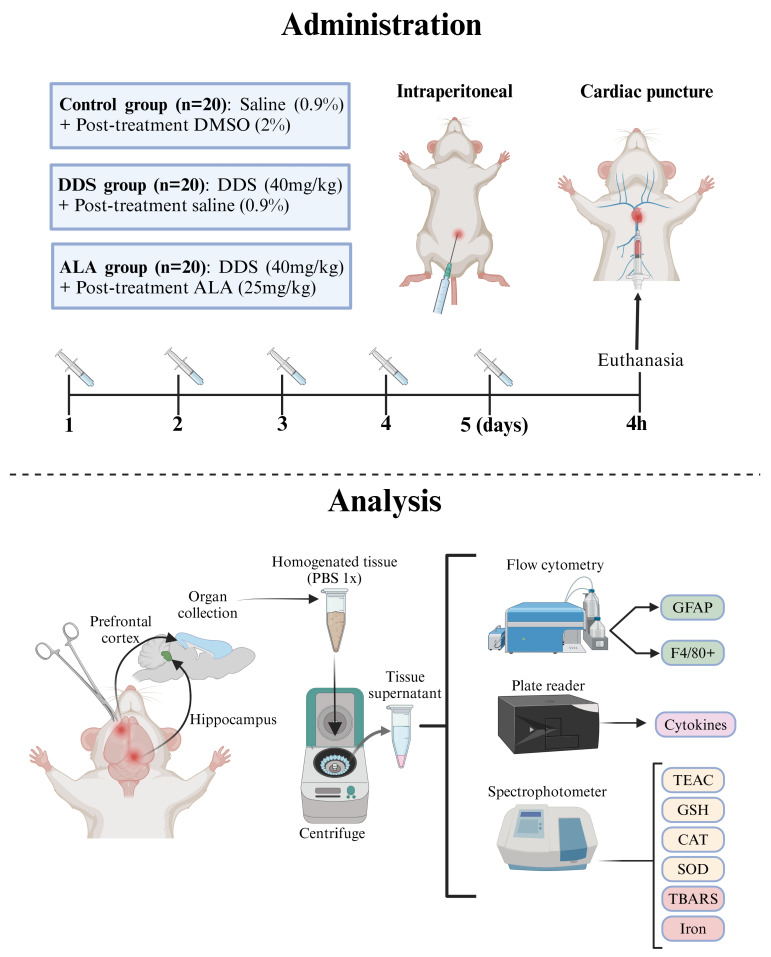
Experimental design of treatment in animals.

**Figure 2 nutrients-17-00791-f002:**
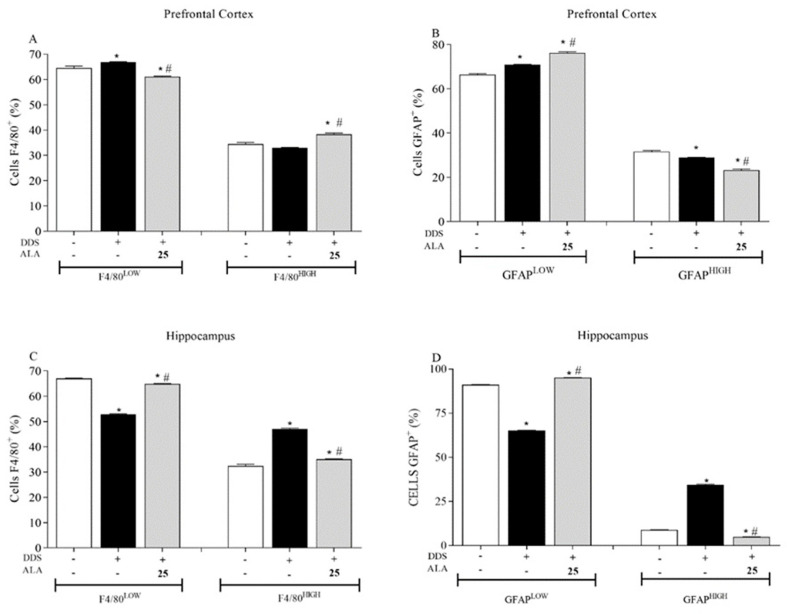
Effects of dapsone (40 mg/kg; ip) on macrophages/microglia (F4/80+) and astrocytes (GFAP+) in the prefrontal cortex (PFC) and hippocampus of mice and post-treatment with ALA (25 mg/kg) for 5 consecutive days. (**A**,**B**) Percentage of macrophages/microglia (F4/80+) and astrocytes (GFAP+) expression in the prefrontal cortex, analyzed by flow cytometry. (**C**,**D**) Percentage of macrophages/microglia (F4/80+) and astrocytes (GFAP+) expression in the hippocampus, analyzed by flow cytometry. Data are presented as mean ± standard deviation (SD). An asterisk (*) indicates a significant effect as determined by two-way repeated measures, ANOVA with Tukey’s test (*p* < 0.05), in comparison with control group. A pound sign (#) indicates a significant effect in comparison with DDS group (*p* < 0.05).

**Figure 3 nutrients-17-00791-f003:**
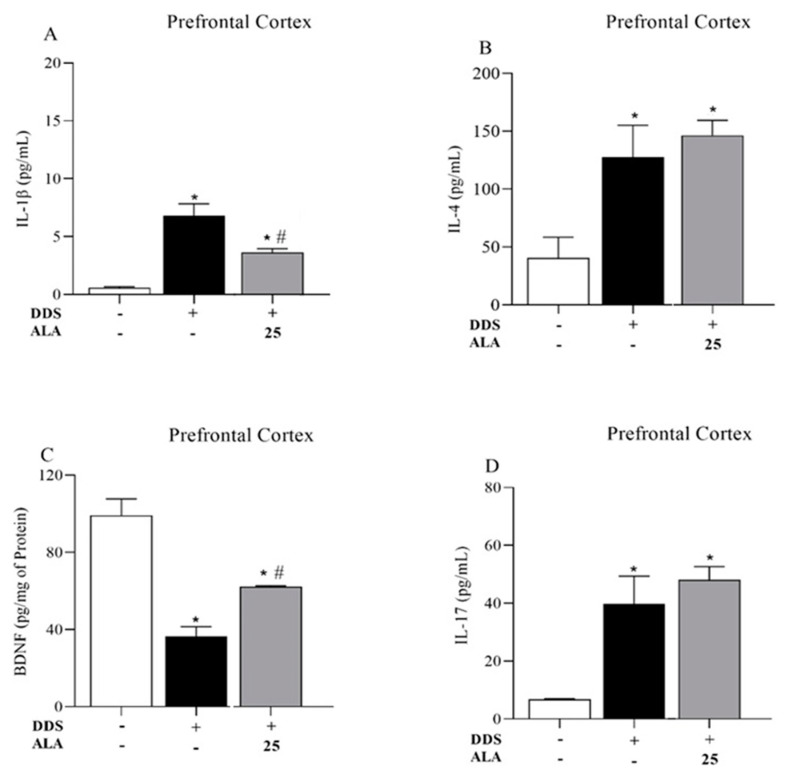
Effects of dapsone (40 mg/kg; ip) on the levels of cytokines IL-1β, IL-17 and IL-4 and BDNF in the prefrontal cortex (PFC) of mice and post-treatment with ALA (25 mg/kg) for 5 consecutive days. (**A**) Concentration of IL1-β (pg/mL), (**B**) concentration of IL-4 (pg/mL), (**C**) concentration of BDNF (pg/mg of protein) and (**D**) concentration of IL-17 (pg/mL) in the prefrontal cortex by ELISA. Data are presented as mean ± standard deviation (SD). An asterisk (*) indicates a significant effect as determined by two-way repeated measures, ANOVA with Tukey’s test (*p* < 0.05), in comparison with control group. A pound sign (#) indicates a significant effect in comparison with DDS group (*p* < 0.05).

**Figure 4 nutrients-17-00791-f004:**
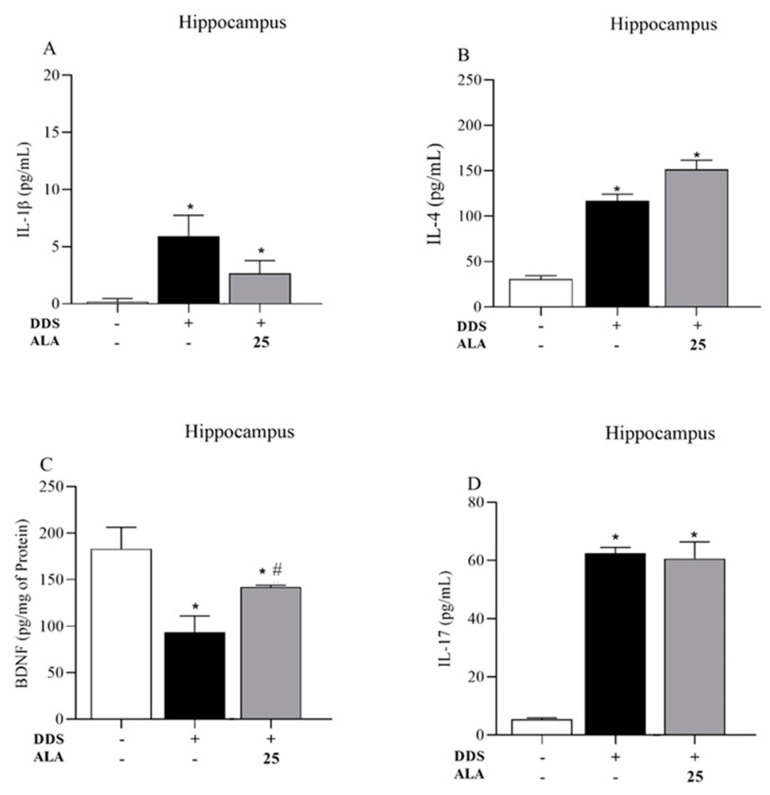
Effects of dapsone (40 mg/kg; ip) on the levels of cytokines IL-1β, IL-17 and IL-4 and BDNF in the hippocampus of mice and post-treatment with ALA (25 mg/kg) for 5 consecutive days. (**A**) Concentration of IL1-β (pg/mL), (**B**) concentration of IL-4 (pg/mL), (**C**) concentration of BDNF (pg/mg of protein) and (**D**) concentration of IL-17 (pg/mL) in the hippocampus by ELISA. Data are presented as mean ± standard deviation (SD). An asterisk (*) indicates a significant effect as determined by two-way repeated measures, ANOVA with Tukey’s test (*p* < 0.05), in comparison with control group. A pound sign (#) indicates a significant effect in comparison with DDS group (*p* < 0.05).

**Figure 5 nutrients-17-00791-f005:**
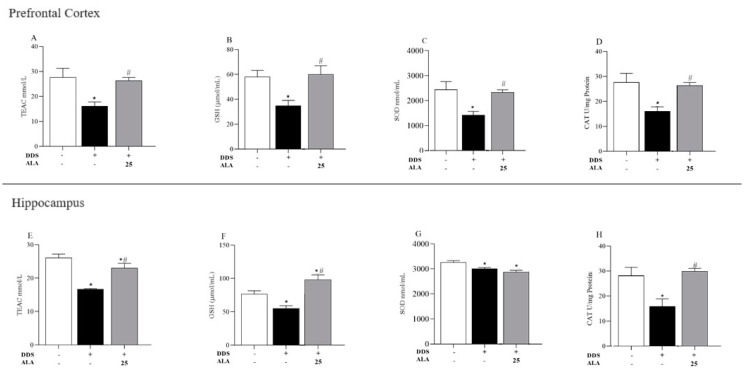
Effects of dapsone (40 mg/kg; ip) on TEAC, GSH, SOD and CAT in the prefrontal cortex and hippocampus of mice and post-treatment with ALA (25 mg/kg) for 5 consecutive days. (**A**) Concentration of TEAC (mmol/L), (**B**) concentration of GSH (µmol/mL), (**C**) concentration of SOD (nmol/mL), (**D**) concentration of CAT (U/mg of protein) in prefrontal cortex by spectrophotometry. (**E**) Concentration of TEAC (mmol/L), (**F**) concentration of GSH (µmol/mL), (**G**) concentration of SOD (nmol/mL), (**H**) concentration of CAT (U/mg of protein) in the hippocampus by spectrophotometry. Data are presented as mean ± standard deviation (SD). An asterisk (*) indicates a significant effect as determined by two-way repeated measures, ANOVA with Tukey’s test (*p* < 0.05), in comparison with control group. A pound sign (#) indicates a significant effect in comparison with DDS group (*p* < 0.05).

**Figure 6 nutrients-17-00791-f006:**
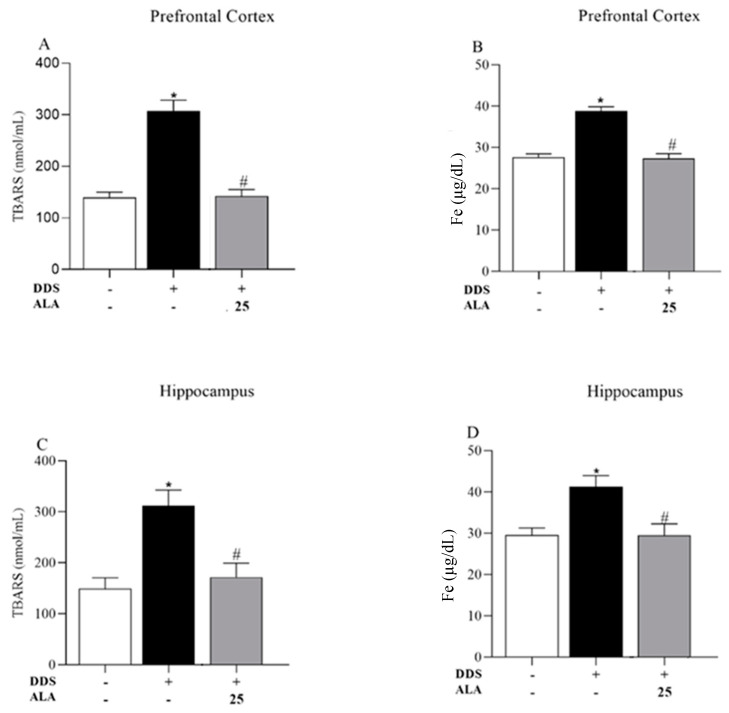
Effects of dapsone (40 mg/kg) on the generation of thiobarbituric acid reactive substances (TBARS) and iron concentration in the prefrontal cortex (PFC) and hippocampus of mice and post-treatment with ALA (25 mg/kg) for 5 consecutive days. (**A**) Concentration of TBARS (nmol/mL), (**B**) concentration of Fe (µg/dL) in prefrontal cortex by spectrophotometry. (**C**) Concentration of TBARS (nmol/mL), (**D**) concentration of Fe (µg/dL) in the hippocampus by spectrophotometry. Data are presented as mean ± standard deviation (SD). An asterisk (*) indicates a significant effect as determined by two-way repeated measures, ANOVA with Tukey’s test (*p* < 0.05), in comparison with control group. A pound sign (#) indicates a significant effect in comparison with DDS group (*p* < 0.05).

## Data Availability

The original contributions presented in this study are included in the article. Further inquiries can be directed to the corresponding author.

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
