# Peer review of "Alpha-Lipoic Acid Reduces Neuroinflammation and Oxidative Stress Induced by Dapsone in an Animal Model"

_nutrients, 2025, doi:10.3390/nu17050791_

Round 1
Reviewer 1 Report
Comments and Suggestions for Authors
The manuscript by Quadros Gomes et al, titled: " Alpha-lipoic Acid Reduces Neuroinflammation and Oxidative Stress Induced by Dapsone in an Animal Model" reported ALA is able to blunt the oxidative and inflammatory damages following Dapsone administration in mice hippocampus. Although the topic is very fascinating, some concerns should be addressed:
- in the abstract the literature gap is missed, and also the aim of the study should be better reported; - In the methods section how did the authors select the ALA dosage? - in the methods section and in results describe what F4/80+ low or high means and what both represents - similar for GFAP: better describe the analysis - All figure legends must be improved: panels name and description are missed - The authors mentioned in discussion several cytokines that are not analyzed in the manuscript: how did the authors select these cytokines? A wide analysis of cytokines should be helpful - The authors assessed the anti-inflammatory effect of ALA, even if the IL-4 levels were increased by DDS and ALA increased IL-17. The authors should better discuss these results - The conclusion reported in line 362 and other are highly speculative and did not supported by the results, thus I suggest to remove the sentences
Author Response
Comments 1: “- in the abstract the literature gap is missed, and also the aim of the study should be better reported.”
Response 1: We agree with this comment. Thefore, we have modified the information in abstract. See page 1, lines 22 to 30.
Comments 2: “- In the methods section how did the authors select the ALA dosage? “
Response 2: Thank you for the question. The dose of 25mg/kg of ALA was chosen based on the methodology described by Sharma and Kupta (2003) and Cremer et al (2006), with adaptations in dose and treatment time, to evaluate its effects on acute exposure and toxic doses of DDS. See page 18, paragraph 4, lines 126 to 129. Furthermore, we previous carried out experiments with a dose of 50mg/kg (Data still not published), which did not show a significant difference in relation to the dose of 25mg/kg. Therefore, we opted for the lowest dose, to avoid possible toxic or pro-oxidant effects of ALA.
Comments 3: “- in the methods section and in results describe what F4/80+ low or high means and what both represents - similar for GFAP: better describe the analysis.”
Response 3: Thank you for your comment. The information was improved in the method (See page 6, paragraph 1, lines 186 to 206). In results, we also include the following information: “High and Low represent the expression of the cellular marker, allowing for a quantitative analysis of cells. In the astrocytes and microglia evaluated in our study, it was also possible to estimate the percentage of cells present, as well as those that are activated or inactive in the tissue and express the marker. See page 7, paragraph 2, lines 237 to 240.
Comments 4: “- All figure legends must be improved: panels name and description are missed.”
Response 4: We agree with this comment. Thefore, we have modified the information in the graphs. See legends in figures 2 to 6.
Comments 5: “- The authors mentioned in discussion several cytokines that are not analyzed in the manuscript: how did the authors select these cytokines? A wide analysis of cytokines should be helpful. “
Response 5: Thank you for your comment. We agree with your point of view. Some information about cytokines was not directly related to this research and was removed. See page 13, paragraphs 2 and 5. However, some studies we believe are important in this discussion on some cytokines, because they involve the effect of ALA or DDS metabolites, which suggests that ALA can act directly or indirectly in several signaling pathways in neuroinflammation or modulation of the immune system.
Comments 6: “- The authors assessed the anti-inflammatory effect of ALA, even if the IL-4 levels were increased by DDS and ALA increased IL-17. The authors should better discuss these results. “
Response 6: Thank you for your comment. We found few studies in the literature on the effects of ALA on the expression or immunomodulation of IL-4 and IL-17, which, to some extent, limited the discussion. Furthermore, regarding these two cytokines, ALA exhibited an anti-inflammatory effect only on IL-4 in the hippocampus.
Comments 7: “The conclusion reported in line 362 and other are highly speculative and did not supported by the results, thus I suggest to remove the sentences.”
Response 7: We agree with this comment. Thefore, we remove the sentence. See page 13, paragraph 5.
Reviewer 2 Report
Comments and Suggestions for Authors
The manuscript by Chagas Monteiro and coworkers deals with a topic that is certainly not new: the role of alpha lipoic acid on neuroinflammation. Nonetheless, the model being investigated is novel and concerns neuroinflammation induced in rodents by DDS.
Major criticism
1) The rationale of this study resides in some phrases in the introduction:
“In addition, degenerative disorders in the central nervous system (CNS), such as Huntington's disease, Parkinson and Alzheimer's disease, can be associated with the prolonged use of DDS, which are caused by iron accumulation, due to disorders in its metabolism, factor that can contribute to the process of oxidative stress, inducing inflammation and neuronal damage [3-5]. “
Reference 3 by Diaz-Ruiz is a review focused on the anti-inflammatory activity of DDS also into the brain (“In recent years its antioxidant, antiexcitotoxic, and antiapoptotic effects have been described in different ischemic damage models, traumatic damage, and models of neurodegenerative diseases, such as Parkinson's (PD) and Alzheimer's diseases (AD).”
Reference 4 by Ward et al. describe the role of iron in brain ageing and neurodegenerative disorders, but never mentions the DDS.
Reference 5 by Carrocci and coworkers again describe the involvement of iron in oxidative stress and neurodegeneration, but never mentions the DDS.
So, for what I can read in the manuscript and in the cited literature, the idea that DDS is a drugs inducing Iron overload into the brain and thus neurodegeneration, remain elusive and unproven.
2) Since the DDS-induced neurodegenerative model seems new, how the dose of 40mg/Kg was chosen? Why not 30 or not 50 mg/Kg? Why intraperitoneally and not orally? This rationale is totally lacking in the manuscript.
3) The same rationale is lacking for ALA dosage. Why 25 mg/Kg? Using 12 mice per group, a number more than sufficient to demonstrate some effects of the treatments, with the same number of animals it would have been possible to test more dosages of ALA to understand whether the effect is dose-dependent or not.
4) Author state that DDS neurodegeneration is associated with prolonged use of DDS (see point 1 above). Are 5 days a good model of a “prolonged” use?
5) Figure 2 results were obtained by flow cytometry, as stated in the legend of Figure 2. But flow cytometry methodology is totally absent in the manuscript. Moreover, also observing Figure 1, how were cell collected for flow cytometry? Authors describe that they started from tissue homogenates. How can be performed flow cytometry starting from tissue homogenates?
6) Statistical issues: Authors write that they started with 20 animals for group (a huge number for our Ethical Committee). But the number of replicates in individual experiments is never highlighted.
Author Response
Comments 1: “In addition, degenerative disorders in the central nervous system (CNS), such as Huntington's disease, Parkinson and Alzheimer's disease, can be associated with the prolonged use of DDS, which are caused by iron accumulation, due to disorders in its metabolism, factor that can contribute to the process of oxidative stress, inducing inflammation and neuronal damage [3-5]. “
Response 1: We agree with this comment. Thefore, we have modified the information in paragraph, based on the articles. See page 2, paragraph 1, lines 51 to 54.
Comments 2: “ Since the DDS-induced neurodegenerative model seems new, how the dose of 40mg/Kg was chosen? Why not 30 or not 50 mg/Kg? Why intraperitoneally and not orally? This rationale is totally lacking in the manuscript.”
Response 2: This dose of 40mg/kg was used based on the methodology described by Bergamaschi et al (2011), with adaptations [75]. See page 3, paragraph 4, lines 124 to 126.
Comments 3: “The same rationale is lacking for ALA dosage. Why 25 mg/Kg? Using 12 mice per group, a number more than sufficient to demonstrate some effects of the treatments, with the same number of animals it would have been possible to test more dosages of ALA to understand whether the effect is dose-dependent or not.”
Response 3: The dose of 25mg/kg of ALA was chosen based on the methodology described by Sharma and Kupta (2003) and Cremer et al (2006), with adaptations in dose and treatment time, to evaluate its effects on acute exposure and toxic doses of DDS. Furthermore, we previous carried out experiments with a dose of 50mg/kg (Data still not published), which did not show a significant difference in relation to the dose of 25mg/kg. Therefore, we opted for the lowest dose, to avoid possible toxic or pro-oxidant effects of ALA.
Comments 4: “Author state that DDS neurodegeneration is associated with prolonged use of DDS (see point 1 above). Are 5 days a good model of a “prolonged” use?”.
Response 4: We believe that this adapted model with 5 days of exposure to high doses of DDS provides us with relevant information about the toxic effects that DDS-NOH and MADDS-NOH, metabolites that probably cause neuroinflammation and oxidative stress in the CNS. At the same time, ALA has shown promoting effects in reducing oxidative changes and neuroinflammation. In addition, Molinelli et al. (2019) report that adverse effects, such as methemoglobinemia and hemolysis, may occur after the second week of treatment with DDS due to the reactivity of its metabolites. However, the chronology of events in humans undergoing regular treatment allows us to estimate that, in animals treated with DDS, considering the proportionality of their lifespan, the same signs could appear from the fifth day of treatment in mice. Finally, we recognize that more detailed studies may be conducted in the future to assess these specific changes, including clinical, laboratory, and even behavioral evaluations. However, the focus of our study was to assess oxidative stress parameters, which are commonly associated with DDS metabolites, as well as the inflammatory profile of cytokines and markers of cellular reactivity in neuronal tissue.
Comments 5: “Figure 2 results were obtained by flow cytometry, as stated in the legend of Figure 2. But flow cytometry methodology is totally absent in the manuscript. Moreover, also observing Figure 1, how were cell collected for flow cytometry? Authors describe that they started from tissue homogenates. How can be performed flow cytometry starting from tissue homogenates? “
Response 5: Thank you for your comment. We have revised the flow cytometry methodology. See page 6, paragraph 1, lines 186 to 206.
Comments 6: “Statistical issues: Authors write that they started with 20 animals for group (a huge number for our Ethical Committee). But the number of replicates in individual experiments is never highlighted.”
Response 6: Thank you for your comment. The information has been improved in the method. 10 animals from each group were used for flow cytometry and 10 were used for the evaluation of cytokines and oxidative parameters. See page 3, paragraph 5, lines 134 to 136.
Reviewer 3 Report
Comments and Suggestions for Authors
The paper by Monteiro and coworkers describes the ameliorating effect of alpha-lipoic acid (ALA) on the oxidative and neuroinflammatory changes in the prefrontal cortex (PFC) and hippocampus of Swiss mice induced by dapsone (DDS). Although, strictly speaking, I am not a specialist in this particular field, to the extent of my understanding, the experimental study is properly done and the conclusions are reasonable. The differences of the effects of DDS and ALA on the PFC and hippocampus observed in this study are very interesting.
There are a few minor points that need to be addressed by the authors before publication.
1. Line 137: Superoxide anion is not “•O2,” but “•O2-.” (“2” is subscript and “- (minus)” is superscript)
2. All over the manuscript: Please insert a half-width space between the value and unit (except “%”). For example, “25°C” (line 136) and “550nm” (line 138) should be “25 °C” and “550 nm,” respectively.
3. Line 143: The “2” in “H2O2” should be subscripted.
Author Response
Comments 1: Line 137: Superoxide anion is not “•O2,” but “•O2-.” (“2” is subscript and “- (minus)” is superscript).
Response 1: We agree with this comment. Thefore, we have made the requested change (O2•-) and highlighted it in red in the manuscript. See page 5, paragraph 1, line 158.
Comments 2: All over the manuscript: Please insert a half-width space between the value and unit (except “%”). For example, “25°C” (line 136) and “550nm” (line 138) should be “25 °C” and “550 nm,” respectively.
Response 2: We agree with this comment. Thefore, we have made the requested change (25 °C and 550 nm) and highlighted it in red in the manuscript. See page 5, paragraph 1, lines 157 and 159, respectively.
Comments 3: Line 143: The “2” in “H2O2” should be subscripted.
Response 3: We agree with this comment. Thefore, we have made the requested change (H2O2) and highlighted it in red in the manuscript. See page 5, paragraph 2, line 165.
Round 2
Reviewer 2 Report
Comments and Suggestions for Authors
In this version, I find the manuscript much improved and therefore acceptable for publication.